# DeepSmile: Anomaly Detection Software for Facial Movement Assessment

**DOI:** 10.3390/diagnostics13020254

**Published:** 2023-01-10

**Authors:** Eder A. Rodríguez Martínez, Olga Polezhaeva, Félix Marcellin, Émilien Colin, Lisa Boyaval, François-Régis Sarhan, Stéphanie Dakpé

**Affiliations:** 1UR 7516 Laboratory CHIMERE, University of Picardie Jules Verne, 80039 Amiens, France; 2Institut Faire Faces, 80000 Amiens, France; 3Faculty of Odontology, University of Reims Champagne-Ardenne, 51097 Reims, France; 4Maxillofacial Surgery, CHU Amiens-Picardie, 80000 Amiens, France; 5Physiotherapy School, CHU Amiens-Picardie, 80000 Amiens, France

**Keywords:** anomaly detection, deep learning, long-short term memory, facial paralysis

## Abstract

Facial movements are crucial for human interaction because they provide relevant information on verbal and non-verbal communication and social interactions. From a clinical point of view, the analysis of facial movements is important for diagnosis, follow-up, drug therapy, and surgical treatment. Current methods of assessing facial palsy are either (i) objective but inaccurate, (ii) subjective and, thus, depending on the clinician’s level of experience, or (iii) based on static data. To address the aforementioned problems, we implemented a deep learning algorithm to assess facial movements during smiling. Such a model was trained on a dataset that contains healthy smiles only following an anomaly detection strategy. Generally speaking, the degree of anomaly is computed by comparing the model’s suggested healthy smile with the person’s actual smile. The experimentation showed that the model successfully computed a high degree of anomaly when assessing the patients’ smiles. Furthermore, a graphical user interface was developed to test its practical usage in a clinical routine. In conclusion, we present a deep learning model, implemented on open-source software, designed to help clinicians to assess facial movements.

## 1. Introduction

According to Jones et al. [1], the human face is an important social stimulus since it provides relevant information about the observed person’s age [2] and sex [2,3]. Moreover, facial expressions are responsible for conveying emotional messages, enhancing communication, and establishing links between individuals [4].

From a clinical point of view, the analysis of facial movements is relevant for diagnosis, care, and follow-up. Primarily, this analysis provides quantitative criteria that ensure an efficient follow-up for patients with facial pathology, e.g., facial paralysis [5]. Several techniques for assessing facial movement have been developed [6], with a view to quantifying the extent of facial paralysis and facilitating diagnosis and therapy, e.g., plastic or reconstructive surgery. Generally speaking, techniques for assessing facial movement can be categorized as either subjective or objective [7].

Subjective assessment techniques are based on the observation made by experienced clinicians; examples include the House–Brackmann facial nerve grading system [8], the Yanagihara facial nerve grading system [9] and the Sunnybrook facial grading system [10]. These methods rely on the graded observation of specific movements. Hence, the method’s level of repeatability can be criticized [11,12,13,14].

Objective techniques are based on the use of sensors to quantify and assess facial movements. Most of these techniques can be automated to some degree, to make them quicker to administer and to reduce variations. However, the signals generated by sensors can be difficult to interpret. Objective assessment techniques can be subdivided into four groups: electromyography (EMG), computer vision, three-dimensional (3D) imaging, and optical motion capture.

The EMG technique consists of measuring the muscle electric response to nerve stimulation. This technique requires one to puncture the small needles of the sensor into a specific facial muscle [15]. Once the sensor is placed, the patient is asked to exercise the muscle to read the electric signals. A non-invasive alternative to EMG is the surface EMG [16], which uses patches of electrodes instead of needles. However, uniformly placing the patches is a hard practice, and the signal is sensitive to external interference [7]. Electroneuronography is another alternative to EMG that consists of comparing distal facial muscle response to electrical impulses. By applying electrical stimulation to the facial nerve trunk, this method records compound muscle action potential as electric signals [17]. Nevertheless, some studies have invalidated distal muscle comparison when assessing facial palsy [18].

Computer vision techniques can be further divided into sparse and dense techniques. The former one leverages on face recognition to automatically place virtual landmarks on the image [19]. Then, some machine learning techniques, such as an ensemble of regression trees [20], or support vector machines [21], are used to classify different levels of facial paralysis based on asymmetry features. These techniques are capable to assess facial palsy [22,23] and social perception [1]. When dealing with the sequence of images, dense techniques are based on optical flow, which describes the face movement in the image space [24,25,26]. Otherwise, when dealing with single images, the assessment can be defined as a classification task, performed by a Convolutional Neural Network (CNN) [27], where asymmetry is used as a feature [28]. However, these techniques use each pixel in the image to predict or classify variants of facial palsy. Although computer vision techniques are faster than the other objective techniques, they are inaccurate since they rely on metric estimations defined on the image space [26].

In practice, the analysis through 3D scans [6] can be used for planning future maxillofacial surgery [29], soft tissues changes quantification [30] and facial mimic variations of patients before and after treatment [31]. This class can be further subdivided depending on the sensor: laser-based scanning, stereophotogrammetry, structured-light scanning, or RGB-D (red, green, blue-depth) sensors [32]. Depending on the sensor, this technique can provide dense information (RGB-D) or sparse information (stereophotogrammetry) if landmarks are placed on the face [33]. Although the stereophotogrammetric technique is the most accurate and reliable, its cost, size, and complexity, are often unsuitable for incorporation into clinical environments with limited availability of resources. One alternative solution to these disadvantages is the use of RGB-D sensors since they can collect accurate static and dynamic 3D facial scans; however, further improvements prior to their implementation are required [32]. In conclusion, the main drawbacks of 3D scan techniques are that most of them do not measure the motion of the face but rather a single 3D model [34], and some of their evaluations rely on subjective analysis.

Optical motion capture techniques use photogrammetry to track the movement of markers in 3D over time. The markers are placed in relevant zones in the face to measure the movement of the skin [35,36].

In [37], a statistical analysis is carried out to assess the presence of unilateral facial palsy before and after surgery. The analysis consists of comparing the trajectories of each pair of distal markers to measure their symmetry. One of the main advantages of motion capture systems is their precision which depends on the camera’s configuration. However, current methods to assess facial movement rely on models that do not fully exploit sequential data [38,39].

Here, we present a deep learning model that assesses facial movement by exploiting optical motion capture data. Compared with the study in [39], we decided to use a deep learning model to represent sequential data collected from healthy movement rather than using a statistical model. Furthermore, we implemented the model through a graphical user interface (GUI), available at https://github.com/PolezhaevaOlga/Face_Motion_Capture (accessed on 12 December 2022), to test its practical usage in clinical routines. Generally speaking, the software evaluates the movement of the patient’s lower third of the face when smiling and provides relevant information on possible diagnoses. Moreover, the software provides a global degree of anomaly that further complements the clinician’s diagnostic. Specifically, the evaluation is carried out with a long short-term memory (LSTM) model [40] defined as an anomaly detector; thus, it compares the patient’s movement with its own prediction. When evaluating a smile, the model is able to predict a healthy estimation of it because it was trained on a dataset that contained solely smiles from healthy volunteers following a one-class anomaly detection strategy [41]. The main objective of this study is to present a different approach, through a deep learning model, to objectively assess facial movement. The second objective is to provide the open-source software, containing the trained model, that served as the proof of concept of our approach.

Below, Section 2 covers the data acquisition and preprocessing as well as the mathematical definitions of the baseline model and our proposed deep learning model. Then, Section 3 details the training and evaluation of the proposed model, the comparison between the latter model and the baseline, and the GUI built to assist clinicians. Lastly, the model’s advantages and limitations are discussed in Section 4, and the paper provides a brief conclusion in Section 5.

## 2. Materials and Methods

This Section focuses on depicting the processes of data acquisition and preprocessing, as well as the models to be compared. The pipeline’s various steps are depicted in Figure 1. Firstly, the data are acquired from motion capture sessions. Secondly, five preprocessing steps are applied to the data. Thirdly, the preprocessed data are used to generate the dataset. Lastly, the dataset is used to compute, train, and evaluate the models.

### 2.1. Motion Capture Sessions

The data used in this paper were acquired in several motion capture sessions. In each session, we recorded a neutral expression and five facial movements: gentle closure of the eyelids, forced closure of the eyelids, pronunciation of the [o] sound, pronunciation of the [pμ] sound and broad smiling. The movements were recorded by tracking 105 reflective markers with an optical-passive motion capture system (Vicon Ltd., Oxford, UK). Prior to the sessions, a group of volunteers and patients were recruited. On the one hand, all volunteers were Caucasian men and women, between 18 and 30 years old, with no facial pathology known. On the other hand, patients were Caucasian men and women with facial pathology. For each volunteer and each patient, a 3D model of the face was generated using a stereo photogrammetry technique (Vectra M3 Imaging System, Canfield Scientific, Parsippany, NJ, USA). Later, a perforated mask was 3D printed (Form 2, Formlabs, Somerville, MA, USA) so that the markers could be placed precisely on the face, using hypoallergenic glue. Moreover, rigid dental support made by a professional prosthetic defined the head’s reference frame to disregard the head movements. During the session, each volunteer and each patient were asked to perform the 5 facial movements. Lastly, the movements were exported in a comma-separated value (csv) file format. The protocol used in the current study was approved by the Local Independent Ethics Committee (CPP Nord-Ouest II, Amiens, France) under references ID-RCB 2011-A00532-39/2011-23 and ID-RCB 2016-A00716-45/2016-55, registered at ClinicalTrials.gov (NCT02002572 and NCT03115203), and performed in accordance with the ethical standards of the 1964 Helsinki Declaration and its subsequent revisions. All participants provided written informed consent for study participation. For further details of the data acquisition process, please refer to [42].

Although the 5 movements previously described were recorded for each participant, this study focuses on the broad smile movement, as its production leads to large muscle displacements. For this purpose, 52 markers of the lower third of the face were chosen (highlighted in Figure 2) to be analyzed further. This consideration was implemented to reduce the complexity of the anomaly detection task, given the small number of csv samples on the dataset.

### 2.2. Data Preprocessing

The motion capture system tracked the 3D ${}^{W}P_{j}\left(t\right)\in R^{3}$ of the *j*-th marker, being j=1,2,…,52, over time in the world coordinate frame W. The dental support is tracked to define the position and orientation of the head reference frame H in the Special Orthogonal group SO(3). The start and the end of each smile are selected manually, and ${}^{W}P\left(t\right)$ was linearly interpolated, so the number of timesteps |t|=400 remained constant.

A homogeneous transformation
(1)$${}^{H}P^{\prime }={}^{H}M_{W}{}^{W}P^{\prime },$$
where $P^{\prime }$ is the homogeneous coordinate of P and ${}^{H}M_{W}$ is the transformation matrix, is applied to express P in H, i.e., ${}^{H}P$. This transformation disregards the head’s translation and rotation, so the face’s movements are precisely tracked.

Much as in [38,39,43], the markers’ displacement was chosen as the feature vector that describes the smile. The markers’ displacement, from their initial position to the current one, was defined by
(2)$$D\left(P_{j}\left(t\right)\right)=∥P_{j}\left(0\right)-P_{j}\left(t\right)∥,$$
where $P_{j}\left(0\right)$ and $P_{j}\left(t\right)$ are, respectively, the initial and current position of the *j*-th marker and D∈R.

Once *D* had been obtained, the *missing values* were estimated using linear regression [44]. However, this regression can be applied only if a small number of values are missing. In other cases, the csv file was not considered. Lastly, a scale transformation (also known as the min-max transformation) was applied to Equation (Equation 2) to normalize the feature vector:(3)$$S\left(D\left(P_{j}\right)\right)=D\left(P_{j}\right)/\max\left(D\left(P\right),D\left(P_{j}\right)\right).$$

### 2.3. Datasets

Firstly, the training and validation datasets, which represent 70% of the smiles, were generated from healthy smiles only (n=25 and n=7, respectively). These datasets were used to compute the baseline and train the deep learning model. Then, the test dataset was generated from 4 healthy smiles (H-test) and 9 patients’ smiles (P-test). The latter smiles were produced by 3 facial palsy patients and 1 facial transplantation patients. This dataset, which represents the 30% of the smiles, is used to evaluate both models. It is important to notice that the latter smiles were produced by volunteers that suffer from facial palsy. Specifically, all the healthy smiles samples were collected from 2014 to 2017 as previously described in [35,36,43,45,46,47].

### 2.4. Models

Two models are considered to evaluate the facial movement of healthy and pathological smiles: the baseline and the LSTM model. In brief, we compare the traditional approach [39], based on a statistical model, with a more complex model. Specifically, our LSTM model was a multivariate time series forecaster that follows a *seq-to-vector* [48] architecture. Although other deep learning models, such as multi-layer perceptron (MLP) [49], CNN, recurrent neural network (RNN) [50], and LSTM-CNN [51], were evaluated for this task, we decided only to include LSTM because its prediction better fitted the healthy smiles on the dataset. Similarly to [52], LSTM showed to be the best anomaly detector using sequential data by computing lower errors than other deep learning models.

#### 2.4.1. Baseline

The baseline model computes a single smile as the average of the markers’ scaled displacement
(4)$$B\left(P\right)=\frac{1}{n}\sum S\left(D\left(P\right)\right),$$
where *n* is the number of smiles (n=32, see Section 2.3). In other words, the baseline can be roughly interpreted as the average smile of the training and validation datasets. Figure 3 shows two examples of the markers’ average scaled displacement.

#### 2.4.2. Long-Short Term Memory

An LSTM model was chosen to predict the smile because it had given a good level of performance for expression recognition by leveraging on sequential data [53]. Moreover, this model has outperformed other deep learning models as a dynamic and time-variant anomaly detector [52].

The LSTM cell [54], displayed in Figure 4, is defined as follows:(5)$$\begin{array}{cc} i_{\left(t\right)} & =\sigma (W_{xi}^{⊤}x_{\left(t\right)}+W_{hi}^{⊤}h_{\left(t-1\right)}+b_{i})\\ f_{\left(t\right)} & =\sigma (W_{xf}^{⊤}x_{\left(t\right)}+W_{hf}^{⊤}h_{\left(t-1\right)}+b_{f})\\ o_{\left(t\right)} & =\sigma (W_{xo}^{⊤}x_{\left(t\right)}+W_{ho}^{⊤}h_{\left(t-1\right)}+b_{o})\\ g_{\left(t\right)} & =\tanh(W_{xg}^{⊤}x_{\left(t\right)}+W_{hg}^{⊤}h_{\left(t-1\right)}+b_{g})\\ c_{\left(t\right)} & =f_{\left(t\right)}⊗c_{\left(t-1\right)}+i_{\left(t\right)}⊗g_{\left(t\right)}\\ y_{\left(t\right)} & =h_{\left(t\right)}=o_{\left(t\right)}⊗\tanh\left(c_{\left(t\right)}\right) \end{array}$$
where g is the gate, f, i, and o are the controller of the forget, input, and output gates, respectively, h is the hidden state, c is the cell, x is the feature vector and
(6)$$\begin{array}{cc} \sigma (z) & ={\left(1+e^{-z}\right)}^{-1}\\ \tanh(z) & =\frac{e^{z}-e^{-z}}{e^{z}+e^{-z}} \end{array}$$
are the logistic sigmoid and hyperbolic tangent functions, respectively, with z∈R.

In the view of the time series forecaster architecture, the mean square error (MSE):(7)$$\mathrm{MSE}=\sum \frac{{\left(\hat{y}-y\right)}^{2}}{n},$$
where y and $\hat{y}$ are the real and the predicted outputs, respectively, were chosen as the *loss function*. Similarly to [55], a windowing process was applied to x. This process consists of generating a batch of *inputs* and *targets* by sliding a window through a vector. The size of the window, defined by window_size=input_size+target_size, was experimentally set to input_size=10 and target_size=1, which results in window_size=11. Thus, resulting on an input size $n_{x}=\left(|j\right|\times \mathrm{window}\_\mathrm{size}\times \left|t|\right)=\left(52\times 11\times 400\right)$.

During training, the model’s parameters were randomly initialized and then updated using the Adam algorithm [56] and the *stochastic gradient descent* method. Several learning rates, batch sizes, number of hidden units and number of hidden layers were experimentally tuned using the Keras Tuner library and the RandomSearch Tuner class for a maximum of 100 epochs with an early stop on the validation loss.

As an anomaly detector, the overall objective of the model was to minimize the degree of anomaly (cost) when evaluating the healthy smiles during training. To this end, the root mean square error (RMSE)
(8)$$\mathrm{RMSE}=\sqrt{\sum \frac{{\left(\hat{y}-y\right)}^{2}}{n}},$$
was selected as the cost function.

## 3. Results

This section presents the results of three experiments: (i) LSTM’s training and evaluation, (ii) facial movement assessments comparison between LSTM and baseline, and (iii) LSTM model deployment on clinician diagnosis through a GUI. All the processes carried out in this study were executed on the same computer whose technical specifications are detailed in Table 1.

The goal of the training is to find the parameters that minimize the cost (or degree of anomaly) on the training dataset. The training’s performance of the LSTM model is shown in Figure 5. The model achieved a performance of 0.0268, 0.0469, 0.0372, 0.0685, for the training, validation, H-test, and P-test datasets, respectively. Some LSTM model’s predictions on the right and left commissure markers, for the H-test and P-Test, are presented in Figure 6.

The LSTM model’s standard deviations were: 0.0119, 0.0274, 0.0174, and 0.0362 for the training, validation, H-Test, and P-Test datasets, respectively.

Next, the baseline and the LSTM models evaluate the test datasets (Table 2) by computing the degree of anomaly (Equation (Equation 8)). Figure 7 illustrates the evaluation, carried out by both models, on the left oral commissure marker of a smile that belongs to the H-Test.

In the view of the results presented here, we built a GUI for facial movement assessment to evaluate its deployment in clinical practice. Therefore, we created DeepSmile (Figure 8), which is an open-source software that assesses facial movement during a smile by running the trained LSTM model as an executable file. DeepSmile uses the csv file as the input to provide a report of its facial movement assessment with the following information: the patient’s data; the performance of each marker (much as in Figure 6); a normalized and metric (in mm) degree of anomaly; a discrete indicator of relevant anomaly based on the normalized degree of anomaly relative to a predefined threshold.

To sum up, the DeepSmile’s facial movement assessment can be carried out as follows. Firstly, the markers are placed on the patient’s face (as in [42]) during the motion capture session. Secondly, the patient is asked to smile while the markers’ locations are recorded. Thirdly, the markers are labeled (again as in [42]) and exported as a csv file. Lastly, the csv file is loaded into DeepSmile through a GUI and a report (based on the LSTM model’s evaluation) is generated.

## 4. Discussion

In short, we have addressed a facial movement assessment problem as a non-linear optimization task rather than a classification task [57] to avoid a certain level of subjectivity, ex. the classes being defined by the grades of the House–Brackmann system. Furthermore, an anomaly detector is able to precisely track the degree of anomaly of a patient during follow-up which is expected to decrease over time if the rehabilition is successful. As stated in [52], LSTM models outperform other machine learning algorithms when sequential data is involved in anomaly detection tasks. On the one hand, the LSTM model is trained to minimize the degree of anomaly when evaluating the healthy smiles on the training dataset. Thus, its predictions are adapted to fit healthy smiles (Figure 6) and flag up a higher degree of anomalies for the patients’ smiles. On the other hand, the baseline computes a single smile as the average of the training dataset. Hence, its assessment relies on a reference smile which does not fit other healthy smiles. Consequently, we consider that the LSTM’s evaluations outperform those of the statistical model commonly used in the literature. Figure 7 illustrates an example where the LSTM’s prediction is well adapted to evaluate healthy smiles whereas the baseline is not. Nonetheless, we consider that further investigation on LSTM variants, such as LSTM auto-encoders [58], should be carried out to exploit sequential data collected by motion capture system which might contain some null values.

Similarly to clinical diagnoses, the variation of the LSTM model’s assessment is lower with healthy smiles than with patients’ smiles. When evaluating the H-test, the RMSE was lower for the LSTM than for the baseline. Conversely, when evaluating the P-test, the RMSE was higher for the LSTM model than for the baseline (Table 2). This implies that the LSTM model is more robust than the baseline to differentiate a healthy smile from a pathological one. Furthermore, the degree of anomaly computed by the baseline on H-1 is higher than the corresponding one computed on P-3 (Table 2). In contrast, all the anomalies computed by the LSTM on the H-test are lower than those computed on the P-test. We, therefore, infer that a model that conveys temporal information is more suitable than a reference average healthy smile for assessing facial movements.

Compared with other deep learning models that leverage on qualitative grading system to evaluate facial pathologies ([59,60]), our LSTM model provides an objective assessment that exploits motion capture data. Furthermore, marker positions are more accurate than landmark positions because metric measurements are directly recorded rather than being estimated from the image space ([22]). Nevertheless, the cameras used by computer vision techniques are cheaper, and their installation requires less space than optical motion capture systems. Although the symmetry was not defined as a characteristic feature of healthy facial movement, ex. [22,61], we observed that our model outputs a low degree of anomaly when evaluating symmetric movement. Indeed, *Miller et al.* report that Emotrics computed more asymmetry in facial landmark positions than Auto E-Face when evaluating healthy volunteers [22] whereas our model did not present this phenomenon. It is interesting to note that the model predicted a displacement of similar magnitude when evaluating a patient with right-sided paralysis (Figure 6c); thus, the model computed a significant degree of anomaly on the paralyzed side. However, the model also computed a smaller degree of anomaly on the healthy side of the face. This reflects the phenomenon of compensation of the non-paralyzed side that we observe in clinical practice, but underlines the fact that, depending on the case, this side cannot really be considered as healthy, at least from the point of view of movement ([62,63]). This data objectively underlines the fact that the management of patients with facial paralysis concerns the whole face. One example is the use of botulinum toxin on the non-paralyzed side to induce a more symmetric mimicry ([64]).

In this study, one of the challenges was to train the model on a small number of healthy smiles. To address it, we opted to reduce the complexity of the anomaly detection task by experimentally selecting a small subset of markers. Indeed, the weights related to the selected markers were higher than the rest when training the deep learning model. Another challenge related to our dataset is that, so far, our deep learning model is trained on healthy smiles produced by Caucasian volunteers only. Additionally, we observed in some patients a passive displacement of markers placed on the paralyzed side. Indeed, the markers were attracted in the direction of the non-paralyzed side by the soft tissue traction phenomenon. Therefore, we concluded that the displacement feature might not fully model abnormal movement, and another feature that includes the direction of movement should be considered.

The GUI we have developed calculates the level of anomaly in millimeters as well as in percentage for all the markers selected. It is, therefore, a global indicator of facial mobility, which can be used for the longitudinal follow-up of patients to quantify the evolution of paralysis. On the one hand, the interest of the GUI is to help clinicians to interpret facial movement, measured by a motion capture system, by displaying a global degree of anomaly. On the other hand, this principle leads to data simplification, whereas we could provide an enhanced diagnosis by fully exploiting another feature. In the future, it would, therefore, be interesting to diversify the algorithm so that it produces a group of scores, as in the Sunnybrook score [10], related to defined anatomical areas or particular functions. Moreover, this group of scores could be related to the relevant zones for each of the 5 movements of our complete protocol. Therefore, we must further curate our data, train more deep learning models on the other 4 movements of our motion capture protocol and explore other loss functions [65], meta-heuristic optimization algorithms [66] and features such as the markers’ positions over time. Lastly, other deep learning architectures, such as auto-encoders [67], LSTM auto-encoders, graph neural networks (GNN) [68] or MLP, could be considered for facial movement assessment using motion capture data.

## 5. Conclusions

In this paper, we present an end-to-end deep learning framework to assess facial movement using optical motion capture data. Our deep learning model is able to detect abnormal movements because it was trained to predict healthy smiles via a one-class anomaly detection strategy. Compared with clinician-graded facial palsy evaluations, our novel technique is repeatable, reliable, objective, and not subjected to observer bias or human error; thus, it can further complement the clinician diagnosis. Furthermore, our training was deployed in a clinical environment, through a GUI, and it demonstrated its potential use, thus, validating the proof of concept. Although, further development is required.

## Figures and Tables

**Figure 1 diagnostics-13-00254-f001:**
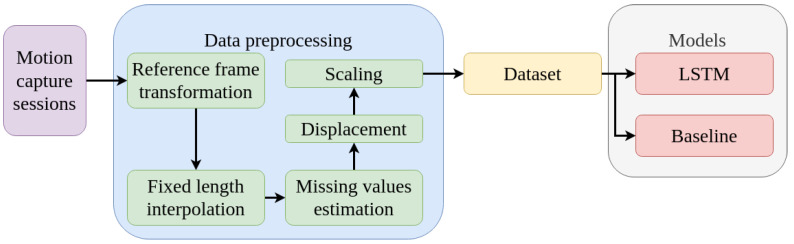
The pipeline’s workflow.

**Figure 2 diagnostics-13-00254-f002:**
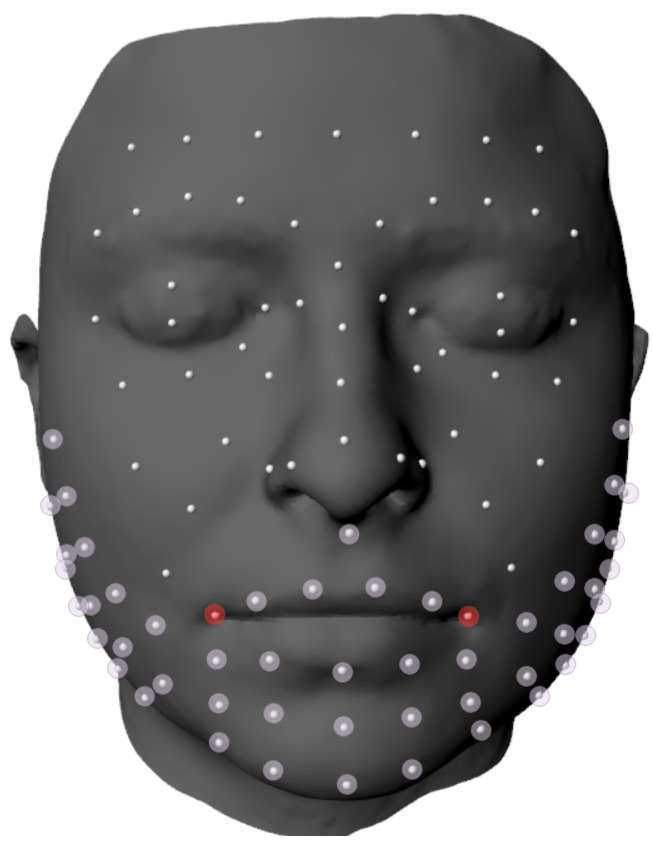
Set of markers virtually placed on a 3D model of the face. The selected markers appear highlighted in violet and red (commissure markers).

**Figure 3 diagnostics-13-00254-f003:**
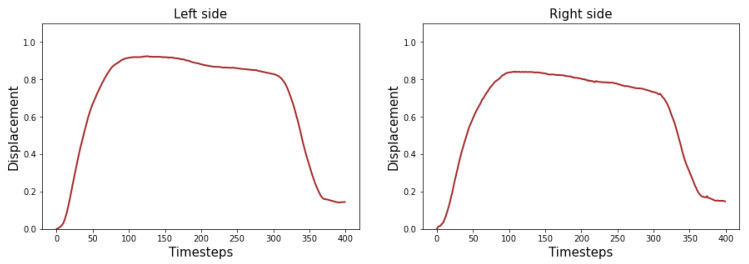
Average of the scaled displacement for the commissure markers.

**Figure 4 diagnostics-13-00254-f004:**
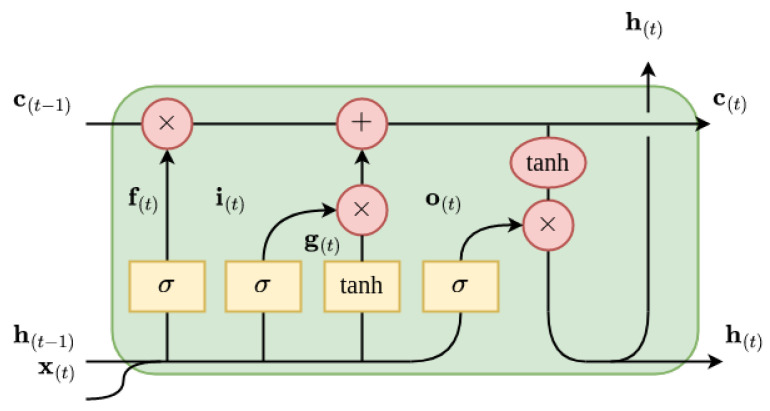
Representation of an LSTM cell.

**Figure 5 diagnostics-13-00254-f005:**
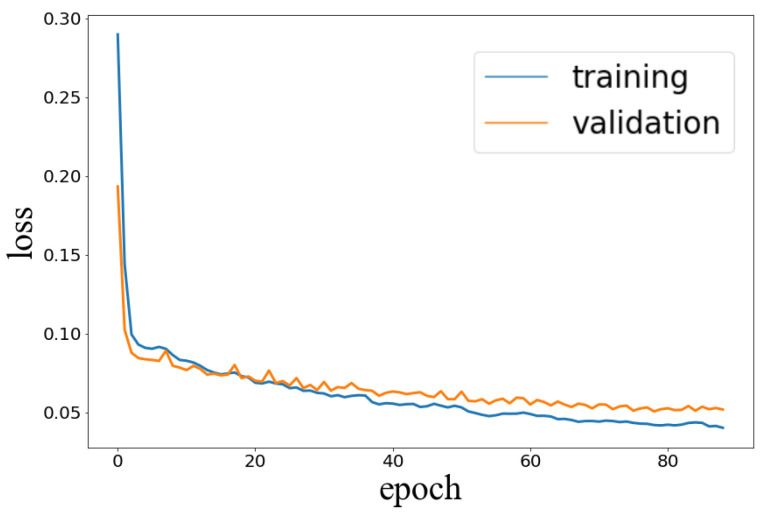
Performance of LSTM during training.

**Figure 6 diagnostics-13-00254-f006:**
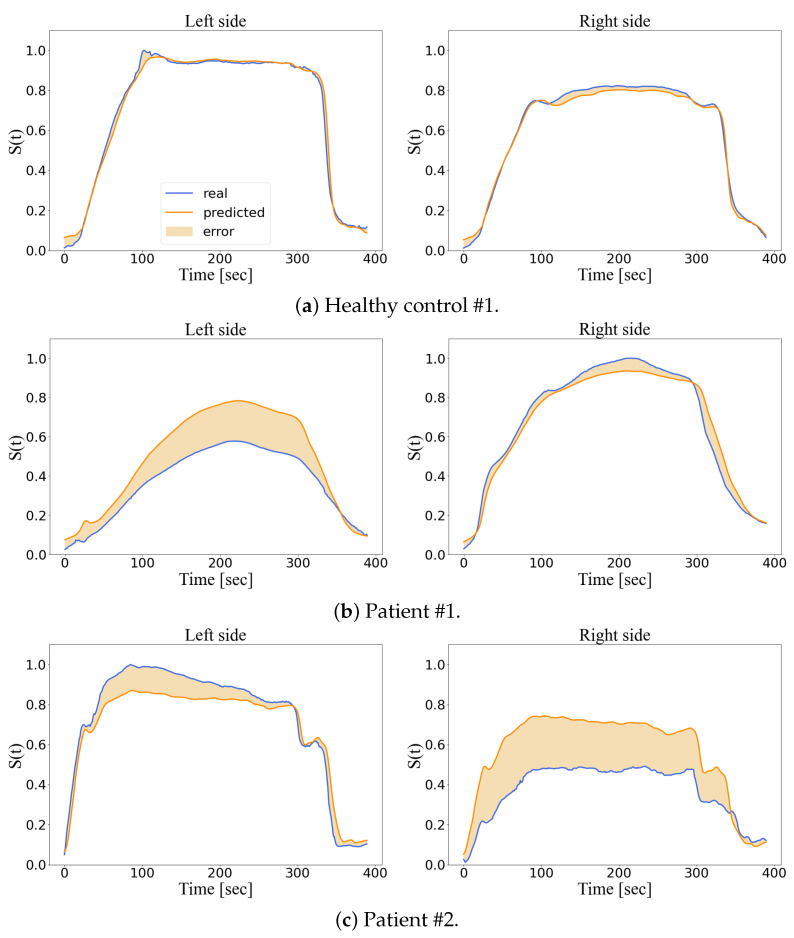
LSTM assessment of a relevant pair of distal markers on the test dataset.

**Figure 7 diagnostics-13-00254-f007:**
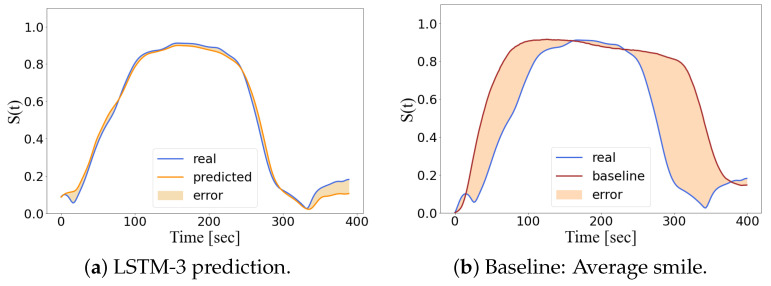
Predictions of the left oral commissure marker on a healthy smile.

**Figure 8 diagnostics-13-00254-f008:**
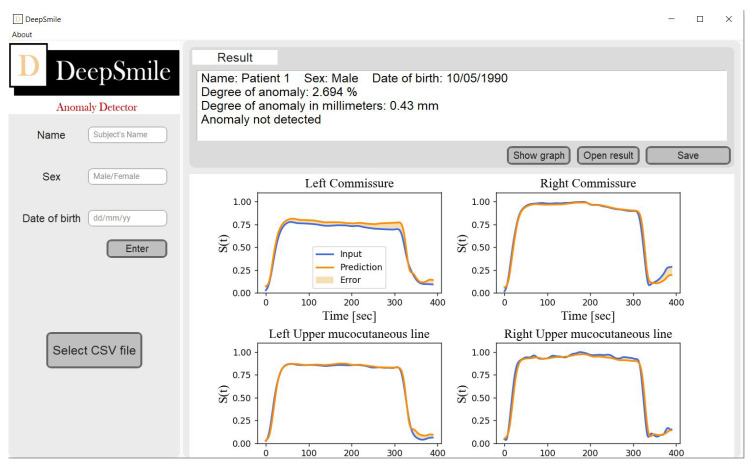
DeepSmile’s graphical user interface.

**Table 1 diagnostics-13-00254-t001:** System specifications.

Hardware or Software	Settings
Model	Asus Strix G15
Operative System	Windows home
GPU	NVIDIA GeForce RTX 3060
Memory (RAM)	16 GB
Processor	AMD Ryzen 7 5800H with Radeon Graphics 3.2 GHz
Memory storage capacity	512 GB SSD
Programming languages	Python 3.10
IDE	Jupyter notebook, Spyder
Libraries	Tensorflow, Pandas, Numpy, Tkinter, Scikit learn

**Table 2 diagnostics-13-00254-t002:** RMSE for the test dataset.

Smile	H-1	H-2	H-3	H-4	P-1	P-2	P-3
LSTM	0.0454	0.0398	0.0527	0.0338	0.0895	0.0976	0.0625
Baseline	0.163	0.103	0.092	0.11	0.168	0.202	0.142

## Data Availability

Code available at https://github.com/PolezhaevaOlga/Face_Motion_Capture (accessed on 12 December 2022).

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
