# Peer review of "DeepSmile: Anomaly Detection Software for Facial Movement Assessment"

_diagnostics, 2023, doi:10.3390/diagnostics13020254_

Round 1

Reviewer 1 Report

The main contributions must be discussed clearly in the introduction section.

Improve the quality of the figures.

Define all variables in the mathematical expressions.

More details of the utilized datasets must be presented.

Caption of the tables must be presented above them not under them.

More comparisons with related studies must be introduced.

Some future directions must be presented.

Reviewer 2 Report

Authors need to address some unsatisfactory points in this article:

- The problem of using pure LSTM: In this problem, the LSTM algorithm is used to learn the motion of the points marked on the face, but neither explained nor reviewed for other deep learning techniques that can do this job, the proposed example is the LSTM autoencoder. Please confirm.

- Using 52 facial markers: Here there is no explanation as to why 52 points are used, and on what basis to choose these 52 points. Why not more, perhaps the more, the higher the accuracy, the more reliable the reliability.

- Ignoring the spatial correlation between points on the face: Maybe using CNN in combination with Timeseries will solve the above problem.

- This article is not rated for reliability. If the problem is classification, we can evaluate the accuracy of the article, but converting to % of the output will make it almost impossible for users to confirm the reliability of the results. It can be understood that 100% is a healthy face, but 90% of a healthy face is called healthy or not and should be reassured or worried.

- Baseline needs to try some other ML methods or a strong method to compare the improvement of the problem.

- Commissure. I do not understand this phrase in the article, that is, the explanation is not thorough, but when giving the concept of left right commissure and related drawings.

- Review on ML is lack of recent works. Authors should provide more comprehensive review on improved version of ML which can help to enhance efficiency of the current work. Authors can find two important examples in [Strengthening Gradient Descent by Sequential Motion Optimization for Deep Neural Networks, https://ieeexplore.ieee.org/abstract/document/9766043].

Round 2

Reviewer 1 Report

It can be accepted

Reviewer 2 Report

Authors have revised carefully the manuscript. The paper is now well-structured and it is suitable for publication.